# Fittings Detection Method Based on Multi-Scale Geometric Transformation and Attention-Masking Mechanism

**DOI:** 10.3390/s23104923

**Published:** 2023-05-19

**Authors:** Ning Wang, Ke Zhang, Jinwei Zhu, Liuqi Zhao, Zhenlin Huang, Xing Wen, Yuheng Zhang, Wenshuo Lou

**Affiliations:** 1Operation and Maintenance Center of Information and Communication, CSG EHV Power Transmission Company, Guangzhou 510000, China; zhujinwei@ehv.csg.cn (J.Z.); zhaoliuqi@ehv.csg.cn (L.Z.); huangzhenlin@ehv.csg.cn (Z.H.); wenxing@ehv.csg.cn (X.W.); zhangyuheng@ehv.csg.cn (Y.Z.); 2Department of Electronic and Communication Engineering, North China Electric Power University, Baoding 071003, China; lws_sure@163.com

**Keywords:** geometric transformation, fittings, object detection, transformer

## Abstract

Overhead transmission lines are important lifelines in power systems, and the research and application of their intelligent patrol technology is one of the key technologies for building smart grids. The main reason for the low detection performance of fittings is the wide range of some fittings’ scale and large geometric changes. In this paper, we propose a fittings detection method based on multi-scale geometric transformation and attention-masking mechanism. Firstly, we design a multi-view geometric transformation enhancement strategy, which models geometric transformation as a combination of multiple homomorphic images to obtain image features from multiple views. Then, we introduce an efficient multiscale feature fusion method to improve the detection performance of the model for targets with different scales. Finally, we introduce an attention-masking mechanism to reduce the computational burden of model-learning multiscale features, thereby further improving model performance. In this paper, experiments have been conducted on different datasets, and the experimental results show that the proposed method greatly improves the detection accuracy of transmission line fittings.

## 1. Introduction

With the development of the economy, the scale of equipment in the power system continues to expand. In order to explore the application prospects and directions of cutting-edge technologies such as artificial intelligence in the field of power, the development of human-machine interaction intelligent systems with reasoning, perception, self training, and learning abilities has become increasingly important research in the field of power [1].

Currently, the length of the power system’s overhead transmission lines has reached 992,000 km and still maintains an annual growth rate of about 5%. Overhead transmission lines are distributed in vast outdoor areas with complex geographical environments, and the traditional manual inspection mode is inefficient [2,3]. In response to the increasingly prominent contradiction between the number of transmission professionals and the continuous growth of equipment scale, the power system promoted the application of unmanned aerial vehicle (UAV) patrol inspection, significantly improving the efficiency of transmission line patrol inspection [4,5,6]. Figure 1 shows patrol inspection images of a transmission line taken by the UAV.

The development of artificial intelligence technology, represented by deep learning, provides theoretical support for the transformation of the overhead transmission line inspection mode from manual inspection to intelligent inspection based on UAV [7]. Object detection is a fundamental task in the field of computer vision. Currently, popular object detection methods mainly use convolutional neural networks (CNN) and Transformer architecture to extract and learn image features. The object detection method based on CNN can be divided into two-stage detection models [8,9,10,11] based on candidate frame generation and single-stage detection models [12,13,14] based on regression. In recent years, the Transformer model for computer vision tasks has been studied by many scholars [15]. Carion et al. [16] proposed the DETR model which uses an encode–decode structured Transformer. Given a fixed set of target sequences, the relationship between the targets and the global context of the image can be inferred, and the final prediction set can be output directly and in parallel, avoiding the manual design. Zhu et al. [17] proposed Deformable DETR, in which the attention module only focuses on a portion of key sampling points around the reference point. With 10× less training epochs, Deformable DETR can achieve better performance than DETR. Roh et al. [18] propose Sparse DETR, which helps the model effectively detect targets by selectively updating only some tokens. Experiments have shown that even with only 10% of encoder tokens, the Sparse DETR can achieve better performance. Fang et al. [19] propose to use only Transformer’s encoder for target detection, further reducing the weight of the Transformer-based target detection model at the expense of target detection accuracy. Song et al. [20] introduce a computationally efficient Transformer decoder that utilizes multiscale features and auxiliary techniques to improve detection performance without increasing too much computational load. Wu et al. [21] proposed an image relative position encoding method for two-dimensional images. This method considers the interaction between direction, distance, query, and relative position encoding in the self-attention mechanism, further improving the performance of target detection.

Applying the object detection models that perform well in the field of general object detection to power component detection has become a hot research topic in the current power field [22,23,24,25]. Zhao et al. [26] use a CNN model with multiple feature extraction methods to represent the status of insulators, and train support vector machines based on these features to detect the status of insulators. Zhao et al. [27] designed an intelligent monitoring system for hazard sources on transmission lines based on deep learning, which can accurately identify hazard sources and ensure the safe operation of the power system. Zhang et al. [28] propose a high-resolution real-time network HRM-CenterNet, which utilizes iterative aggregation of high-resolution feature fusion methods to gradually fuse high-level and low-level information to improve the detection accuracy of fittings in transmission lines. Zhang et al. [29] first proposed that there is a visual indivisibility problem with bolt defects on transmission lines and that the attributes of bolts, such as whether there are pin holes or gaskets, are visually separable. Therefore, bolt recognition is considered a multi-attribute classification problem, and a multi-label classification method is used to obtain accurate bolt multi-attribute information. Lou et al. [30] introduce the position knowledge and attribute knowledge of bolts into the model for the detection of visually indivisible bolt defects, further improving the detection accuracy of visually indivisible bolt defects.

Although there have been some related studies on transmission line fittings detection in the field of electric power, quite difficult problems remain. The main performance is shown in the following aspects: (1) Due to the variable viewing angles of UAV photography, the shape of some fittings varies greatly under different shooting visions, resulting in poor detection performance of fittings under different viewing angles. As shown in Figure 2, the blue frame is the bag-type suspension clamp, and the red frame is the weight. As can be seen from Figure 2, the appearance of the bag-type suspension clamp and the weight has undergone significant changes under different shooting visions. (2) Figure 3 shows the area ratio of different fittings tags in different transmission line datasets. It can be seen that the scale of different fittings in each dataset varies greatly, which is an important factor affecting detection performance. (3) The UAV edge device is small in size and has limited storage and computational resources, so the detection model cannot be too complex. To address the above issues, this paper proposes a transmission line fittings detection method based on multi-scale geometric transformation and attention-masking mechanism (MGA-DETR). The main contributions of this article are as follows:We have designed a multi-view geometric transformation enhancement strategy that models geometric transformations as a combination of multiple homomorphic images to obtain image features from multiple views. At the same time, this paper introduces an efficient multi-scale feature fusion method to improve the detection performance of transmission line fittings from different perspectives and scales.We introduced an attention-masking mechanism to reduce the computational burden of model-learning multiscale features, thereby further improving the detection speed of the model without affecting its detection accuracy.We conducted experiments on three different sets of transmission line fittings detection data, and the experimental results show that the method proposed in this paper can effectively improve the detection accuracy of different scale fittings from different perspectives.

The rest of the paper is organized as follows: Section 2 describes the method proposed in this paper, we propose a multi-view geometric enhancement strategy, introduce an efficient multi-scale feature fusion method, and design an attention-masking mechanism to improve model performance. Section 3 conducted experiments on different datasets and evaluated the methods proposed in this article. Finally, the conclusive remarks are given in Section 4.

## 2. Methods

The fittings detection method based on multi-scale geometric transformation and attention-masking mechanism (MGA-DETR) proposed in this paper is shown in Figure 4. The method is mainly divided into four parts: backbone, encoder, decoder, and prediction head. The backbone is used to extract image features and convert them into one-dimensional image sequences. In the encoder, the self-attention mechanism is used to obtain the relationship between image sequences, and then the trained image sequence features are output. The decoder initializes the object queries vector and is trained by the self-attention mechanism to learn the relationship between the object queries vector and image features. In the prediction header, a binary matching method is used to classify the category of the object queries vector and locate the position of the boundary box, completing the detection of transmission line fittings.

Firstly, we designed a multi-view geometric transformation strategy (MVGT) to improve the detection performance of the model for fittings under different visual conditions in the backbone network part. Then, we introduced an efficient multi-scale feature fusion method (BiFPN) to improve the detection accuracy of the model for objects with different scales. Finally, to reduce the computational complexity of the model and achieve efficient transmission line inspection, this paper introduces an attention-masking mechanism (AMM). This method improves model detection by designing a scoring mechanism to filter out image regions that are less relevant to model detection.

### 2.1. Multi-View Geometric Transformation Strategy

When the distribution of test samples and training samples is different, the performance of object detection will decrease. There are many reasons for this problem, such as changes in the surface of objects under different lighting or weather conditions. Most methods to solve this problem focus on obtaining more data to enrich the feature representation of the object. In the field of object detection, there are usually two ways to obtain richer image feature representations. One method uses models to generate virtual images and add them to the dataset to increase the amount of data [31,32,33]. The other method uses methods such as random clipping and horizontal flipping to obtain high-quality feature representations during data preprocessing [34,35,36]. However, these methods do not pay attention to the geometric changes of the object caused by different shooting angles. This problem is particularly prominent in the inspection of power transmission lines. When the drone is shooting from different angles of view, the appearance of fittings can signifi-cantly change, leading to missed inspections and false inspections. Based on the above reasons, as shown in Figure 5, we propose the MVGT module that uses homomorphic transformation to bridge the gap between objects caused by geometric changes, and then fuses image features to improve the detection performance of fittings at different shooting angles.

The homography transformation is a two-dimensional projection transformation that maps a point in one plane to another plane. Here, a plane refers to a planar surface in a two-dimensional image. The mapping relationship of corresponding points becomes the homography matrix. The calculation method is as follows:(1)$${\left(x_{i},y_{i},w_{i}\right)}^{T}=H_{i}\times {\left(x_{i}{}^{,},y_{i}{}^{,},w_{i}{}^{,}\right)}^{T}$$
where $x_{i},y_{i}$ are the horizontal and vertical coordinates of the original image, and $x_{i}{}^{,},y_{i}{}^{,}$ are the horizontal and axial coordinates of the image after the homography transformation. Set $w_{i}=w_{i}{}^{,}=1$ as the normalization point. $H_{i}$ is a 3 × 3 homography matrix, it can be expressed as follows:(2)$$H_{i}=\left(\begin{array}{l} h_{00}{\;h}_{01}{\;h}_{02}\\ h_{10}\;h_{11}\;h_{12}\\ h_{20}\;h_{21}\;h_{22} \end{array}\right)$$

So the $x_{i}{}^{,}$ and $y_{i}{}^{,}$ can be calculated by the following:(3)$$x_{i}{}^{,}=\frac{h_{00}x+h_{01}y+h_{02}}{h_{20}x+h_{21}y+h_{22}}$$
(4)$$y_{i}{}^{,}=\frac{h_{10}x+h_{11}y+h_{12}}{h_{20}x+h_{21}y+h_{22}}$$

Therefore, when the coordinates of the four corresponding points are known, the homography matrix $H_{i}$ can be obtained. In this paper, we have designed n sets of homography matrices to obtain corresponding homography-transformed images. After that, the homomorphic transformed image features are spliced to obtain features with the size of H×W×NC. Finally, we use a 1×1 convolution pair to reduce the dimension of the fused feature to the H×W×C dimension. By combining the image features after homography transformation, the model can learn pixel changes from different perspectives, further improving the detection performance of fittings in transmission lines.

### 2.2. Bidirectional Feature Pyramid Network

UAVs fly high in the sky with a wide field of vision. The transmission line images captured by UAVs contain multiple categories of fittings. As shown in Figure 2, the range of fittings scales in different datasets are widely distributed. In the inspection of transmission lines, it is often due to the low resolution of small-size fittings, the missing details of the fittings, and the lack of features that can be extracted, which can easily lead to issues such as missing inspection. Therefore, the detection of such fittings has become the focus and difficulty of research.

In object detection methods, feature pyramid networks (FPN) are mainly used to improve the detection ability of models for objects of different scales [37]. As shown in Figure 6a, the main idea of the FPN is to fuse the context information of image features, enhancing the representation ability of shallow feature maps, and improving the detection ability of small-scale objects. Aiming at the defect of only focusing on one direction of information flow in FPN, Liu et al. [38] proposed the PAFPN to further fuse image features of different scales by adding a bottom-up approach, as shown in Figure 6b. In this paper, we introduce a bidirectional feature pyramid network (BiFPN) to optimize multiscale feature fusion in a more intuitive and principled manner [39], as shown in Figure 6c.

First, assume that there is a set of image features $P_{i}\in \left\{P_{1},P_{2},\ldots ,P_{n}\right\}$ with different scales. Where $P_{i}$ represents the image features of the i level resolution. Effective multiscale feature extraction can be considered as a process in which $P_{i}$ fuses different resolution features through a special spatial transformation function, with the ultimate goal of achieving feature enhancement. The fusion process is shown in Figure 6a, in which the network uses image features at levels 3 to 7, with the feature resolution at the level i being $1/2^{i}$ times the input image resolution.

BiFPN adopts a bidirectional feature fusion idea that combines top-down and bottom-up. In the top-down process, the seventh level node is deleted, which only has a single resolution input and has a small contribution to feature multiscale fusion. Deleting this node can simplify the network structure. At the same time, combining a top-down route with a bottom-up route increases the hierarchical resolution information required for the scale fusion process with minimal operational costs. Unlike the FPN, which only performs one feature fusion operation, the BiFPN regards the fusion process as an independent network module, connecting multiple feature fusion modules in series to achieve more possible fusion results.

In the top-down and bottom-up routes, upper and lower sampling methods are used to adjust the size of the feature map to be consistent, and a fast normalized feature fusion algorithm is used to fuse the adjusted feature map. The basic idea of a fast normalized feature fusion algorithm is that each target to be identified has its specificity, such as diverse scales and complex backgrounds. Therefore, visual features of different scales have different contributions to the network detection of the object. This paper uses learnable scalar values to measure the contribution of different levels of resolution features to the final prediction of the network. Using the softmax function to limit scalar values is a good method, but softmax can significantly reduce the GPU processing speed. To achieve acceleration, using a direct normalization algorithm can solve this problem:(5)$$w_{i}{}^{,}=\frac{w_{i}}{\varepsilon +\sum_{j}w_{j}}$$
where ε is a minimum value. In order to avoid numerical instability that may occur during normalization calculations, we set ε=0.0001. The $w_{i}$ is the learned scalar value. To ensure $w_{i}\geq 0$, we use the ReLU activation function for each generated $w_{i}$.

The improved network uses three different scale features $P_{3},\;P_{4}$ and $P_{5}$ extracted from the backbone network as inputs for cross-scale connectivity and weighted feature fusion. Take node $P_{5}$ as an example:(6)P5t−d=Conv(w1⋅P5+w2⋅Resize(P6)w1+w2+ε)
(7)P5b−u=Conv(w1,⋅P5+w2,⋅P5t−d+w3,Resize(P4)w1,+w2,+w3,+ε)
where $P_{5}{}^{t-d}$ is a top-down intermediate feature and $P_{5}{}^{b-u}$ is a bottom-up output feature. Resize is an up-sampling operation or a down-sampling operation. Conv is a convolution operation.

### 2.3. Attention-Masking Mechanism

Although the model can obtain multiscale features of images using the BiFPN, there are still some problems. On the one hand, the self-attention mechanism in DETR can only process one-dimensional sequence data, and images belong to two-dimensional data. Therefore, when processing images, it is necessary to first perform dimensionality-reduction processing on the images. On the other hand, the image for object detection generally has a high resolution and mostly contains multiple targets at the same time. If the image is dimensionally reduced directly, the computational complexity of the Transformer codec will significantly increase. In order to solve this problem, in DETR, CNN is first used to extract image features and simultaneously reduce image dimensions, to control the overall calculation amount within an acceptable range. However, after using the BiFPN, the calculation amount of the model will be multiplied. To solve this problem, this paper introduces an attention-masking mechanism [40]. Firstly, a scoring network is used to predict the importance of the image sequence data input to the encoder, and the image sequence is trimmed hierarchically. Then, an attention-masking mechanism is used to prevent attention computation between the trimmed sequence data and other sequence data, thereby improving the computational speed.

The attention-masking mechanism designed in this paper is hierarchical, and as the calculation progresses, image sequence data with lower scores are gradually discarded. Specifically, we set a binary decision mask $S\in {\left\{0,1\right\}}^{N}$ to determine whether to discard or retain relevant data, where N is the length of the image sequence. When S=0, it means that the data need to be discarded, but it is reserved anyway. During training, we initialize all S to 1 and gradually update S as the training progresses. Then, for the image sequence x in the input encoder, it is first passed into the MLP layer to obtain local features:(8)$$f^{local}=MLP(x)$$

Then, we interact with S on the local features of the image sequence to obtain the global features of the current image:(9)$$f^{global}=Agg(MLP(x),S)$$
where A can be obtained by simple averaging pooling:(10)$$Agg(f^{local},S)=\frac{\sum_{i=1}^{N}S_{i}f_{i}{}^{local}}{\sum_{i=1}^{N}S_{i}}$$

Local features contain information about specific data in an image sequence, while global features contain all contextual information about the image. Therefore, we combine the two and transmit them to another MLP layer to obtain the probability of discarding or retaining image sequence data:(11)$$s=Soft\max\left[MLP(f^{local},f^{global})\right]$$

Subsequently, in order to maintain the length of the input image sequence during the training process unchanged while canceling the attention interaction between the trimmed sequence data and the data therein, we designed an attention-masking mechanism (AMM). To put it simply, AMM is added to attention calculation:(12)$$e_{ij}=\frac{(x_{i}w_{Q}){\left(x_{j}w_{K}\right)}^{T}}{\sqrt{d}}$$
(13)$$G_{ij}=\left\{ \begin{array}{l} 1\;i=j\\ 0\;i\neq j \end{array}\right.$$
(14)$$a_{ij}=\frac{\exp(e_{ij})G_{ij}}{\sum_{k=1}^{n}\exp(e_{ik})G_{ij}}$$
where x is the data in the image sequence, $w_{Q}$ and $w_{k}$ are learnable parameter matrix, and d is used for normalization processing.

## 3. Experimental Results and Analysis

We trained the model using AdamW [41], setting the learning rate of the initial Transformer to 0.0001, the learning rate of the backbone network to 0.00001, the weight attenuation to 0.001, and the batch size to 8. The training process adopts the cosine annealing algorithm. When the detection accuracy of the validation set no longer increases, the learning rate is reduced by 10% until the learning rate accuracy no longer increases through adjustment. For the hyperparameter in the experiment, we set the number of object queries vectors to 100, and the number of layers of the Transformer encoder–decoder to 6. The experimental part was implemented using the Python framework and trained and tested using an NVIDIA Geforce GTX Titan device with four GPUs.

### 3.1. The Introduction of Datasets

In recent years, aerial photography technology has grown rapidly. To collect images of transmission lines, an aerial unmanned aerial vehicle (UAV) is not only simple to operate, but also can collect information quickly and safely. We used the UAVs aerial photography technology to obtain a large number of images of power transmission lines. The UAVs are equipped with a high-definition image transmission system, which can capture high-definition images of power transmission lines. Due to the different depth of field in the imaging of transmission line images captured by UAVs, we constructed three datasets in the experiment to verify the performance of the model.

(1) Fittings Datasets-25 (FD-25): Based on the progress of current UAV shooting technology, we constructed the fittings dataset of high-definition transmission line images captured by UAVs at ultra-wide angles. The characteristic of this dataset is that it has a wide shooting range and contains a large number of fittings. We annotated the images according to the MS-COCO 2017 dataset’s annotation specifications. The dataset includes a total of 4380 images and 50,830 annotation boxes. It includes 25 annotation categories, namely triangle yoke plate, right angle hanging board, u-type hanging ring, adjusting board, hanging board, towing board, sub-conductor spacer, shielded ring, grading ring, shock hammer, pre-twisted suspension class, bird nest, glass insulator without coating, compression tension class, suspension class, composite insulator, bowl hanging board, ball hanging ring, yoke plate, weight, extension rod, glass insulator with coating, lc-type yoke plate, upper-level suspension clamp, and interphase spacer. To our knowledge, the Fittings Dataset-25 currently contains the largest number of fittings components in the power industry and has the most detailed classification of fittings categories. An example image of the dataset is shown in Figure 7a,e.

(2) Fittings Datasets-12 (FD-12): In addition to the transmission line images captured by UAVs at ultra-wide angles, we also annotated the relatively close-range transmission line images captured by UAVs. The datasets included 1,586 images and 10,185 annotation boxes. This includes 12 categories of fittings, including pre-twisted suspension clamp, bag-type suspension clamp, shielded ring, grading ring, spacer, wedge-type strain clamp, shockproof hammer, hanging board, weight, parallel groove clamp, u-type hanging ring, and yoke plate. Compared to the Fittings Datasets-25, the Fittings Datasets-12 has shorter shooting distances, fewer types of fittings, and a relatively rough classification of fittings. The image of the datasets is shown in Figure 7b,f.

(3) Fittings Datasets-9 (FD-9): There are a considerable number of small-scale fittings in transmission lines. Taking bolts as an example, the proportion of bolts in transmission line images is very small; usually, only a few pixel sizes; which leads to low accuracy of bolt recognition in object detection models. In response to the above issues, this paper cropped the Fittings Datasets-25 and Fittings Datasets-12, saving the areas with more small-scale fittings as new images and annotating them to increase the proportion of small-scale fittings in the input images. The dataset includes 1,800 images and 18,034 annotation boxes. This includes nine types of fittings: bolt, pre-twisted suspension clamp, u-type hanging ring, hanging board, adjusting board, bowl head hanging board, bag-type suspension clamp, yoke plate, and weight. An example image of the dataset is shown in Figure 7c,g.

### 3.2. Comparative Experiment

To verify the effectiveness of the proposed method in the fittings detection of transmission lines, we first conducted experiments using different models in the datasets constructed in this paper. As shown in Table 1, the AP is the average accuracy of the model detecting all labels in the datasets. GFLOPs are Giga Floating point Operations Per Second, FPS is the number of frames transmitted per second, and params is the number of parameters for the model.

From Table 1, it can be seen that in the three types of fittings datasets, the MGA-DETR proposed in this paper achieves the highest average precision (AP) in fittings-detecting transmission lines. In the fittings datasets-9, the AP of MGA-DETR reached 88.7%, an increase of 3.1% compared to the baseline model DETR. In the fittings datasets-12, the AP value of MGA-DETR reached 83.4%, an increase of 4.8% compared to the baseline model DETR. In fittings datasets-25, the AP value of MGA-DETR reached 66.8%, an increase of 5.1% compared to the baseline model DETR. Compared to the three types of datasets, the detection accuracy of the fittings datasets-25 is relatively low because the images in the dataset are taken at ultra-wide angles, and the same image contains a variety of fittings types with significant scale changes. Through experiments, it has been proven that the model proposed in this article is of great help for the fittings detection of transmission lines. Comparing the params of different models, it can be found that the YOLOX has the smallest params. YOLOX is a single-stage object detection model. YOLOX introduces anchor-free, greatly reducing computational complexity while avoiding anchor-parameter tuning. Therefore, it has relatively large advantages in GFLOPs, FPS, and params. The method proposed in this paper is based on the transformer, and due to the self-attention mechanism in the transformer, the computational complexity of the model is relatively large. Compared to other methods based on transformer, our method introduces AMM, which successfully accelerates the calculation speed of the model and reduces the number of parameters in it. The MGA-DETR proposed in this paper has improved the params and FPS of the Deformable DETR, which also uses FPN, further proving the effectiveness of the proposed method.

Figure 8 shows the detection performance of the proposed method in different fittings datasets. Among them, Figure 8a,d show the detection performance of Fittings Datasets-25, Figure 8b,e show the detection performance of Fit tings Datasets-12, and Figure 8c,f show the detection performance of Fittings Datasets-9. From the figure, it can be seen that the method proposed in this article effectively detects the presence of fittings in the image in all three types of datasets. Taking Figure 8b,e as examples, the shape of the bag-type suspension clamp in the image has undergone significant changes due to different shooting angles. However, the method in this paper accurately detects two different shapes of bag-type suspension clamps. This further proves the effectiveness of the MAGT module proposed in this paper.

Table 2 shows the detection results of fittings at different scales in three datasets. Among them, the glass insulator, grading ring, and shielded ring are large-scale fittings; the adjusting board, yoke plate, and weight are mesoscale fittings; and the hanging board, bowl hanging board, and u-type hanging ring are small-scale fittings. The × symbols in Table 2 indicate that the dataset does not contain fittings of this category. Through comparison, it can be seen that our proposed MGA-DETR has better performance in fittings detection at different scales. Taking the small-scale fittings hanging board as an example, the AP in three datasets was 86.9%, 80.4%, and 63.1%, respectively. Compared with the baseline model, the DETR increased by 7.2%, 4.5%, and 9.7%, respectively. The experiment shows that the introduction of the BiFPN in DETR has better detection performance for different scales of fittings.

### 3.3. Ablation Experiment

In this section, we designed a series of ablation experiments to demonstrate the effectiveness of each module used in this paper. We used the Fittings Datasets-12 with moderate shooting distance and relatively rich fittings categories to verify the AP of the model.

As shown in Table 3, we analyzed the impact of different module combinations on the experimental results. When all three models are not used, the AP at this time is 78.6%. When only the MVGT module is used, the AP of the model increases by 1.5%, indicating that the feature combination after image homography transformation is beneficial for detecting fittings under different visual conditions. When only the BiFPN is used, the AP of the model increases by 1.8%, indicating that multi-scale feature fusion is more effective in transmission line images with significant scale changes. When only using the AMM module, the AP of the model increases by 1.1%, indicating that the model can improve detection accuracy by filtering out irrelevant background information. When three modules are added simultaneously, the AP reaches its maximum.

In Table 4, we analyzed in detail the impact of different numbers of homography transformations on model performance. When the number is 0, the AP of the model is only 81.7%. With the fusion of image features after homography transformation, the model performance reaches its optimal level at the number of 4, with an AP of 83.4%. When the number of homomorphic transformations further increases, the model performance decreases, indicating that the model has fully learned the geometric transformations in different views at this time. Our analysis concludes that the reason is that with the increase in the number, the model overfitting will lead to a decrease in AP.

As shown in Table 5, we analyzed the impact of different FPNs on model performance. When FPN is not used, the model’s AP is only 81.6%. When using FPN, the AP increased by 0.6%, indicating that learning multi-scale image features helps the model detect transmission line fittings at different scales. However, FPN only considers the top-down feature fusion, while PAFPN considers the bottom-up feature fusion on this basis. However, the efficiency of the two feature-fusion methods did not reach the optimal level. In this article, we introduced the BiFPN, which further improved the AP of the model, demonstrating the effectiveness of our method.

## 4. Conclusions

In order to improve the accuracy of transmission line fittings detection, this paper proposes a fittings detection method based on multi-scale geometric transformation and attention-masking mechanism. Firstly, we designed an MVGT module to utilize homography transformation to obtain image features from different views. Then, the BiFPN was introduced to efficiently fuse multi-scale features of images. Finally, we used an AMM module to improve model speed by masking the attention interaction between image sequence data with lower scores and other data. This paper constructs three different datasets of transmission line fittings and conducts experiments on them. The experimental results show that the proposed method effectively improves the performance of transmission line fittings detection. In the next step of our work, we will study the deployment of the model to obtain its application in the industry.

## Figures and Tables

**Figure 1 sensors-23-04923-f001:**
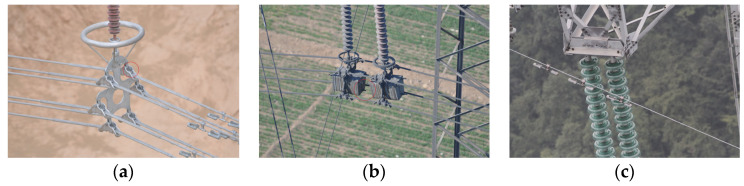
Transmission line images captured by the UAV.

**Figure 2 sensors-23-04923-f002:**
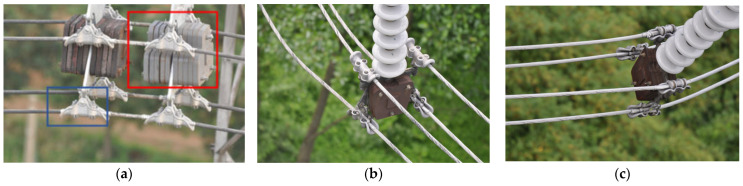
Transmission line images from different shooting angles.

**Figure 3 sensors-23-04923-f003:**
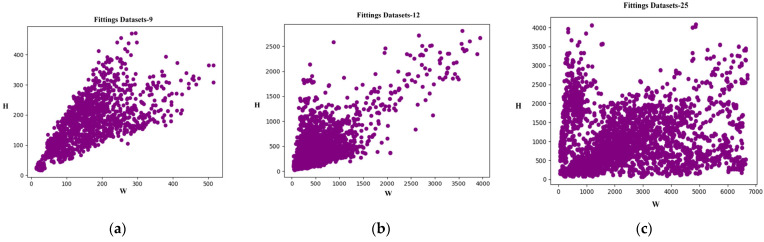
Scale distribution of fittings in different transmission line datasets.

**Figure 4 sensors-23-04923-f004:**
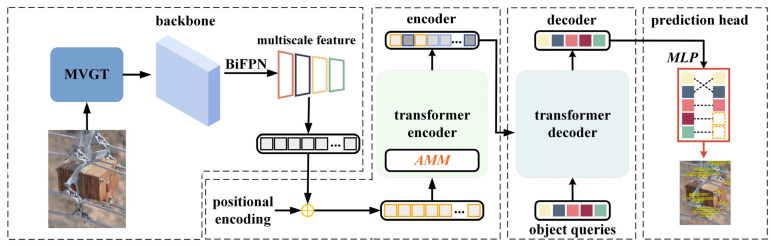
The basic architecture of the MAG-DETR.

**Figure 5 sensors-23-04923-f005:**
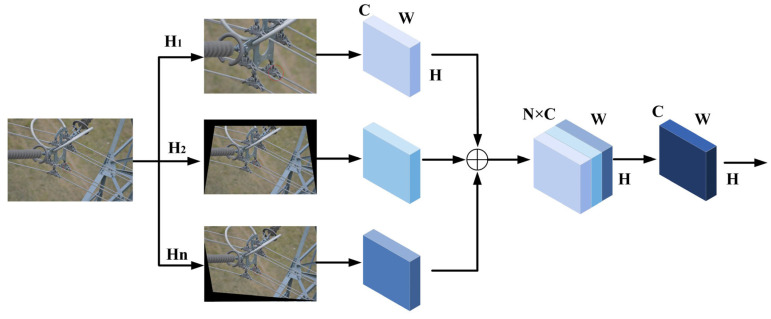
The architecture of the module of MVGT.

**Figure 6 sensors-23-04923-f006:**
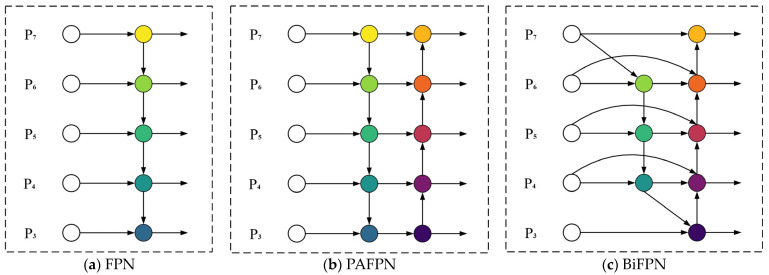
The architectures of different FPNs.

**Figure 7 sensors-23-04923-f007:**
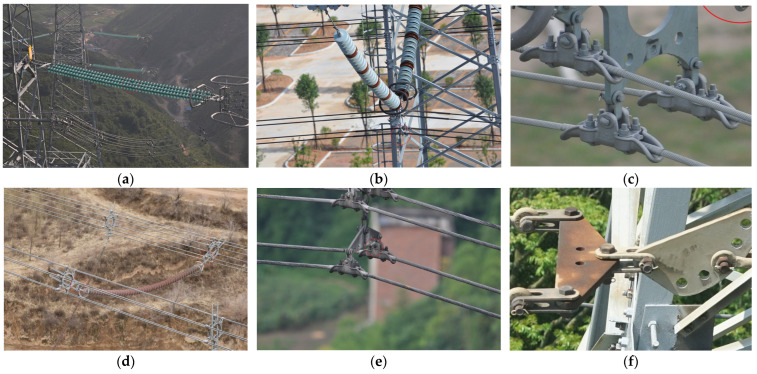
Images from different datasets.

**Figure 8 sensors-23-04923-f008:**
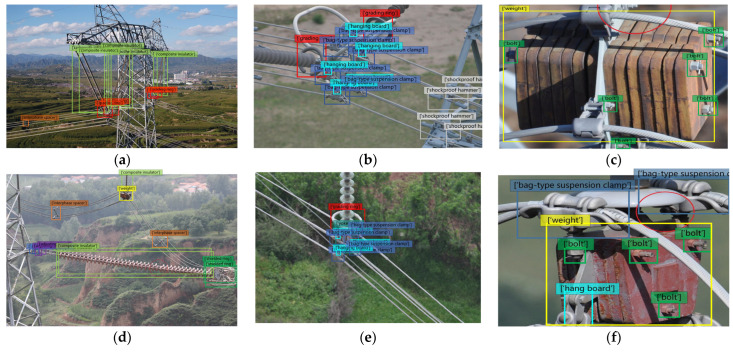
The architectures of different FPN.

**Table 1 sensors-23-04923-t001:** Experimental results of different fittings datasets.

Model	AP (FD-9)	AP (FD-12)	AP (FD-25)	GFLOPs/FPS	Params
Faster R-CNN	80.2	75.1	59.4	246/20	60 M
YOLOX	83.4	78.3	61.3	**73.8/81.3**	**25.3 M**
DETR	85.6	78.6	61.7	86/28	41 M
Deformable DETR	85.9	81.2	62.5	173/19	40 M
Sparse DETR	86.2	81.5	63.2	113/21.2	41 M
**MGA-DETR**	**88.7**	**83.4**	**66.8**	101/25.7	38 M

**Table 2 sensors-23-04923-t002:** Experimental results of DETR/MGA-DETR on different categories in three datasets.

Fittings	AP (FD-9)	AP (FD-12)	AP (FD-25)
glass insulator	×	×	×
grading ring	×	83.1/**89.7**	72.6/**80.4**
shielded ring	×	83.2/**90.2**	69.8/**79.5**
adjusting board	87.3/**90.7**	78.8/**85.1**	57.9/**68.7**
yoke plate	87.9/**91.2**	79.3/**84.4**	58.3/**69.1**
weight	88.2/**91.3**	78.2/**85.2**	57.5/**68.2**
hanging board	79.7/**86.9**	75.9/**80.4**	53.4/**63.1**
bowl hanging board	81.3/**86.6**	76.1/**80.5**	52.7/**62.9**
u-type hanging ring	82.6/**86.9**	75.4/**80.1**	53.5/**62.3**

**Table 3 sensors-23-04923-t003:** The impact of different modules on experimental results.

Model	MVGT	BiFPN	AMM	AP (FD-9)	AP (FD-12)	AP (FD-25)
MGA-DETR	×	×	×	85.6	78.6	61.7
	√	×	×	85.9	80.1	63.2
	×	√	×	86.3	80.4	63.9
	×	×	√	85.8	79.7	62.9
	√	√	×	87.6	82.9	65.4
	√	×	√	87.3	81.6	64.7
	×	√	√	87.4	81.7	64.9
	√	√	√	**88.7**	**83.4**	**66.8**

**Table 4 sensors-23-04923-t004:** The influence of different numbers of homography transformations on experimental results.

Model	Number	AP (FD-9)	AP (FD-12)	AP (FD-25)
MVGT	0	87.4	81.7	64.9
	1	87.8	82.5	65.3
	2	88.0	82.9	65.9
	3	88.3	83.1	66.5
	4	**88.7**	**83.4**	**66.8**
	5	88.6	83.3	66.6
	6	88.1	82.7	66.1

**Table 5 sensors-23-04923-t005:** The Influence of Different FPNs on Experimental Results.

Model	FPN	PAFPN	BiFPN	AP (FD-9)	AP (FD-12)	AP (FD-25)
MGA-DETR	×	×	×	85.1	81.6	60.7
	√	×	×	86.7	82.3	62.1
	×	√	×	87.2	82.8	64.3
	×	×	√	**88.7**	**83.4**	**66.8**

## Data Availability

Not applicable.

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
