# Peer review of "Fittings Detection Method Based on Multi-Scale Geometric Transformation and Attention-Masking Mechanism"

_sensors, 2023, doi:10.3390/s23104923_

Round 1

Reviewer 1 Report

This paper proposes a fittings detection method based on multi-scale geometric transformation and attention masking mechanism. A multi-view geometric transformation enhancement strategy is used to model geometric transformations. An attention masking mechanism is proposed to reduce the computation. Experimental results on a transmission line fittings detection dataset show that the proposed method is able to improve the detection accuracy of different scale fitting.

The proposed algorithm is somewhat novel and meaningful. However, there are still some issues in this paper:

1.     In the section 1. Introduction, the authors indicate “The UAV edge device is small in size and has limited storage and computational resources, so the detection model cannot be too complex.” How to resolve this problem in this paper? Please give more explanation.

2.     In the section 3.3, the authors use datasets-12 to verify the effectiveness of the proposed module in this paper. Why don't they consider all the datasets? 

3.     Datasets-25 have 25 different categories. Why only 9 kinds of fittings in Table 2? Glass insulator should be in Datasets-25. Why dose the AP of glass insulator is “X”?

4.     In order to represent the differences between these three datasets, please provide more images in Figure 7.

5.     How to get the images of these three datasets? Can authors make these datasets public for other researchers to conduct comparative experiments?

 I suggest the authors polish this paper.

Reviewer 2 Report

In general, this is a clear and well-written manuscript. The detection method proposed in the paper is ingenious, and on the basis of the DETR framework, multi-scale geometric transformation and attention masking mechanism are introduced to improve the performance of transmission line fittings detection. The scientific and sufficient experiments show the superior performance of the proposed method. However, there are some problems that should be improved.

1.      In the Comparative Experiment part, the evaluation indexes including AP, GFLOPs/FPS need to be explained more about their physical meaning than just a numerical comparison.

2.      In the table 1, the YOLOX algorithm has the smallest params and the smallest GFLOPs/FPS, the reason for this and the description of this algorithm should be expanded.

3.      In the table 2, the “model” in the first row and column should be replaced with “fitting”. And The categories of the three datasets are not just nine fittings in this table, and why only these nine are compared needs to be explained.

4. Whether is the research supported by any program funding?

No comment.
